# Drug Release Profiles and Disintegration Properties of Pectin Films

**DOI:** 10.3390/ma12030355

**Published:** 2019-01-24

**Authors:** Yoshifumi Murata, Chieko Maida, Kyoko Kofuji

**Affiliations:** Faculty of Pharmaceutical Science, Hokuriku University, Ho-3, Kanagawa-machi, Kanazawa 920-1181, Japan; c-maida@hokuriku-u.ac.jp (C.M.); k-kofuji@hokuriku-u.ac.jp (K.K.)

**Keywords:** pectin, film dosage form, film disintegration, drug release rate

## Abstract

We assessed the disintegration profiles of the film dosage forms (FDs) prepared using pectin by measuring the amount of pectin dissolved from the films in a limited amount of aqueous medium. Furthermore, we used miconazole and dexamethasone as standard drugs and investigated the relationship between the disintegration rate of the FDs and the rate of drug release. We used two types of pectin in this study to develop thin films with a thickness of approximately 25–35 μm. The FDs gradually disintegrated in the aqueous medium, and the disintegration profile of the FDs differed depending on the types of pectin. In addition, the rate of disintegration of the film matrix affected the dissolution rate of the drug incorporated into the FD. Thus, our results show that FDs prepared using pectin are beneficial because of their high solubility in a limited amount of medium, and the rate of drug release from the FDs can be regulated by selecting a specific type of pectin or by altering the concentration of the film base.

## 1. Introduction

Film dosage forms (FDs) are excellent novel dosage forms for targeted drug delivery, because FDs rapidly swell and disintegrate in the bodily fluids and thus release the drug incorporated in the gel matrix of the FD [1,2,3,4,5,6]. The FD is a tool by which drugs can be delivered to local disease sites in the oral cavity [7,8]. Although the drug loading capacity of films is typically very low, FDs are easy to administer, particularly in the case of patients who have difficulty in swallowing the conventional oral dosage forms. The method of preparation of FDs affects the stability of the drug; therefore, a method that does not involve the use of heat is preferred for the development of FDs. FDs have been prepared using some water-soluble polysaccharides as the film base, because these are safe for oral administration, and thin films can be formed using simple methods that do not require dissolution in an organic solvent. FDs prepared using natural polysaccharides such as sodium alginate or pullulan are beneficial for the treatment of localized conditions in the oral cavity [9,10].

Pectin is an anionic water-soluble polysaccharide, which consists of galacturonic acid and its methyl ester. Pectin is safe for oral administration, and thus, it has been used as a food additive, mainly as a gelling agent [11,12,13]. Pectin is also an attractive material for the preparation of pharmaceuticals [14,15,16,17]. In addition, pectin is used as a coating agent for films, because of the ease of coating using pectin.

Here, we prepared a solution of pectin as the film base without dissolving it in an organic solvent or heating. Then, the FD was obtained by evaporation of the solvent; the FD was prepared using pectin dissolved in an aqueous medium; therefore, the disintegration rate of the film matrix is an important factor when characterizing FDs. Previously, we have described a simple colorimetric assay to measure the amount of pectin in an aqueous solution [18]. The disintegration profiles of the films were assessed by measuring the amount of pectin from each FD that was dissolved in the test medium. In addition, we incorporated miconazole nitrate (MCZ) or dexamethasone (DM) into the FDs as model drugs, and we investigated the dissolution profiles of these drugs from the FDs in a limited amount of aqueous medium.

## 2. Materials and Methods

### 2.1. Materials

We used six types of pectin as film bases; apple pectin (A-PT) and citrus pectin (C-PT) were purchased from SIGMA (St. Louis, MO, USA), and the four other types of pectins (LM-5CS-J, LM-12CG-J, LM-18CG, and X-602-03) were obtained from CP Kelco (Tokyo, Japan). MCZ, DM, and hydroxylamine (HX) were purchased from Wako Pure Chemical Industries, Ltd. (Osaka, Japan). A water-soluble carbodiimide, 1-cyclohexyl-3-(2-morpholinoethyl) carbodiimide metho-p-toluenesulfonate (CMEC), was purchased from Aldrich Chemical Co. (Milwaukee, WI, USA). All other chemicals were of reagent grade and were obtained from commercial sources.

### 2.2. Viscosity of Film Base Solution

We dispersed 2–4% (w/w) pectin in deionized water to prepare the film base solution. The solution was stirred for 1 day at room temperature, and then the viscosity was measured at 25 °C using a viscometer (VM-1G-M; CBC Materials, Tokyo, Japan).

### 2.3. Preparation of FDs

The FDs were prepared as follows: the film base solution was thoroughly mixed by sonication and poured (3 g each) into individual plastic petri dishes (diameter, 54 mm). The dishes were kept for 24 h at 37 °C; then, the circular films formed were transferred into a desiccator. In addition, the FD containing the model drug was prepared using the pectin solution in the same manner. The thickness of the film was measured at 10 points on each film using a micrometer (CLM1-15QM; Mitutoyo, Kawasaki, Japan) at a pressure of 0.5 N. We measured the thickness of three films and calculated the mean thickness.

### 2.4. Film Disintegration Test

A film was placed in a plastic dish (diameter, 54 mm) and 10 mL physiological saline preheated to 37 °C was added. Then, the dish was then shaken (300 rpm) in an incubator (SI-300; As One Co., Osaka, Japan) set at 37 °C. The medium (0.3 mL) was periodically removed using a plastic syringe and filtered through a syringe-driven filter unit (pore size, 0.45 µm). An equal volume (0.3 mL) of physiological saline at 37 °C was added to the dish in the incubator to maintain a constant volume. Aliquots (0.2 mL) of the filtered solution were added to 0.8 mL of ion-exchanged water in test tubes, and then, the solution was mixed well using a vortex mixer. The amount of pectin in each sample solution (1 mL) was measured using the method described below. Each test was performed in triplicate.

### 2.5. Colorimetric Assay of Pectin

We used 20 mM HX in ion-exchanged water and 0.1 M CMEC in 2% pyridine-HCl buffer (pH 5.0). We added 1-mL aliquots of HX and CMEC reagents to 1 mL of the sample solution followed by vortex mixing. Each mixture was incubated at 40 °C for 20 min, and then, 20 mM FeCl_3_ in 0.1 M HCl (3 mL) was added. The absorbance of the solution in a quartz cell (1-cm light path) was measured at 480 nm using a spectrophotometer (UV-1200; Shimadzu, Kyoto, Japan). Each absorbance value was normalized to that of a blank reagent. For each test, a calibration curve was constructed using a fresh set of pectin standards.

### 2.6. Drug Dissolution Test

The sample solution was obtained using the same method as that described in the film disintegration test section. We placed 80-µL aliquots of the filtered sample solution in micro test tubes (1.5 mL); then, we added 720 µL of methanol to precipitate the pectin dissolved from the dosage form. The samples were mixed and centrifuged (7700 *g*, 5 min; H-1300; Kokusan Co., Saitama, Japan), and the supernatants were injected onto the HPLC column. Each test was performed in triplicate.

### 2.7. Assay of Model Drugs

The HPLC system (Hitachi Co., Tokyo, Japan) consisted of a pump (L-2130), a UV-detector (L-2400), an autosampler (L-2200), and a chromate-integrator (D-2500) connected to a packed column (150 mm × 4.6 mm, Cosmosil 5C18-MS-II; Nacalai Tesque Inc., Kyoto, Japan). The detector wavelength was set at 230 nm. To determine the concentration of MCZ, we performed HPLC at ambient temperature with an eluent that consisted of 10 mM KH_2_PO_4_ and acetonitrile (1:4) at a flow rate of 1.0 mL/min [19]. For the assay of DM, the mobile phase consisted of 10 mM phosphate buffer (pH 2.3) and acetonitrile (13:7) at a flow rate of 1.0 mL/min [20].

## 3. Results and Discussion

The base solution was prepared without heating and was poured into a petri dish to form films by using the casting method; therefore, viscosity of the solution was an important factor in the preparation of FDs. The viscosity of each 4% pectin solution is shown in Table 1. The solutions prepared using LM-5CS-J, LM-12CG-J, LM-18CG, and X-602-03 pctins showed low viscosity (26–70 mPa·s), and they could be readily cast. Further, A-PT and C-PT had sufficiently low viscosity for casting. The solution prepared using 5% A-PT had high viscosity and thus films could not be cast using this solution.

To check the film formation by the casting method, we poured 4% pectin into molds and evaporated the solvent from the solution. Circular films were formed when the base solutions prepared using A-PT and C-PT did not contain a model drug, the thicknesses of the films prepared using these solutions was approximately 25–35 μm (Figure 1, Table 2). Although thin films could be formed using the other types of pectin, these films showed cracks. In addition, thin circular films could be formed using A-PT and C-PT solutions containing MCZ (1 mg/g) or DM (0.25 mg/g); therefore, these pectins were selected as the film bases.

The FDs prepared using pectin swelled and disintegrated, which resulted in the release of pectin into the aqueous medium (Figure 2). The FDs prepared using 2% A-PT showed dissolution of 31 ± 4% of the incorporated pectin by 10 min, and about 80% of the film base dissolved into the physiological saline at 30 min. The film disintegration rate gradually decreased with an increase in the concentration of A-PT in the base solution. For example, 20 ± 3% of the incorporated pectin dissolved after 10 min from the FD prepared using 3% A-PT and less than 10% of the pectin dissolved in the case of FDs prepared using 4% A-PT. The FDs prepared using C-PT showed rapid disintegration. The FDs prepared using 2% C-PT showed that approximately 70% of the incorporated pectin dissolved by 10 min, and nearly the entire amount of C-PT contained in the FD dissolved after 30 min (Figure 3).

The drug incorporated in the FD dissolved when the dosage form immersed in the test medium. Our results showed that 1.1 ± 0.3 mg of MCZ dissolved at 10 min from the FD prepared using 2% A-PT, and 1.6 ± 0.4 mg of MCZ dissolved at 30 min (Figure 4). The dissolution rate of MCZ decreased when 4% A-PT was used as the film base solution. Furthermore, the drug release profile from FDs prepared using 4% C-PT was similar that of FDs prepared using 2% A-PT. In addition, we observed that the disintegration profile of the FD affected the rate of drug release from the dosage forms containing DM. Further, 0.42 ± 0.14 mg of DM dissolved by 10 min from the FD prepared using 2% A-PT, and approximately 90% (0.70 mg) of the total DM incorporated in the FD dissolved after 30 min (Figure 5). DM showed a slow rate of dissolution from the FDs prepared using 4% A-PT; 0.43 ± 0.02 mg of DM had dissolved by 30 min from the FDs prepared using 4% A-PT. FDs prepared using 2% A-PT and 4% C-PT as the film base solution showed similar drug release profiles. The disintegration profiles of the FDs differed depending on the type of pectin used as the film base. Moreover, erosion of the FDs affected the dissolution rate, particularly of the water-insoluble model drug DM, from the dosage forms.

## 4. Conclusions

We prepared FDs using different types of pectin as the film base to determine the dissolution of drugs in a limited amount of biological media. Then, we investigated the relationship between the disintegration profiles of the FDs and the rates of drug release from these FDs. Two pectins, A-PT and C-PT were selected as the film base and 2–4% of the base solution were used to prepare FDs considering the viscosity for casting. The FDs gradually disintegrated in the test medium, and the disintegration of the film matrix affected the dissolution rate of the drug incorporated in the FD. Thus, our results show that FDs prepared using pctins are beneficial because of their high solubility in a limited amount of medium. In addition, FDs may be excellent vehicles for oral drug delivery because of their safety and properties that enable controlling the drug released at a specific site that has a limited amount of aqueous medium, such as the oral cavity. Furthermore, we think that it is necessary to investigate the drug release profile from FD in the simulated fluid at the site to be applied, such as an artificial salivary solution.

## Figures and Tables

**Figure 1 materials-12-00355-f001:**
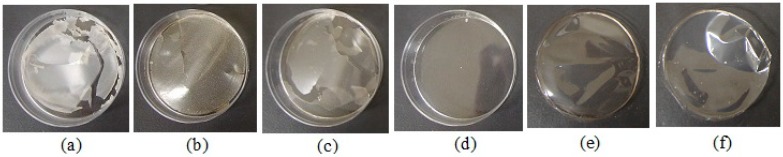
Images of films prepared using 4% pectin (without the drug). (**a**) LM-5CS-J; (**b**) LM-12CG-J; (**c**) LM-18CG; (**d**) X-602-03; (**e**) apple pectin, A-PT; (**f**) citrus pectin, C-PT.

**Figure 2 materials-12-00355-f002:**
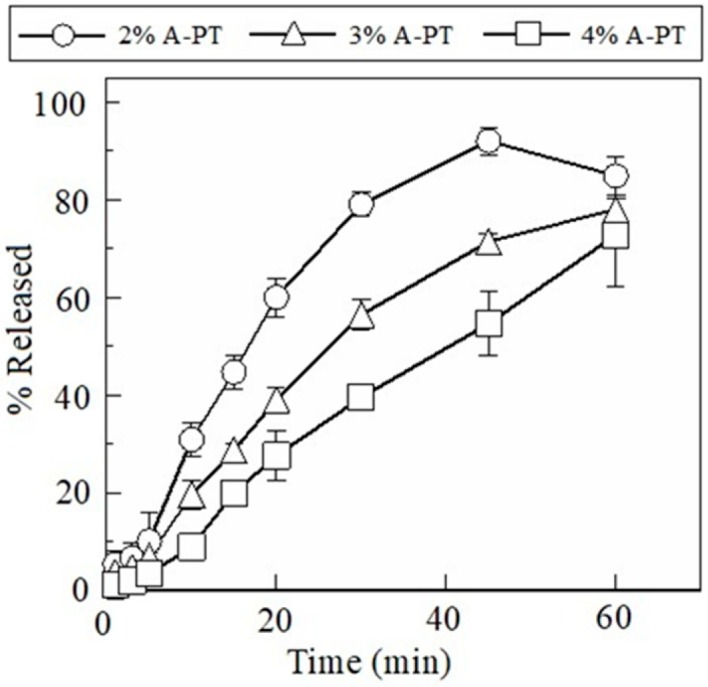
Dissolution profiles of pectin from film dosage forms prepared using apple pectin (A-PT). Each result represents the mean and standard deviation of three determinations.

**Figure 3 materials-12-00355-f003:**
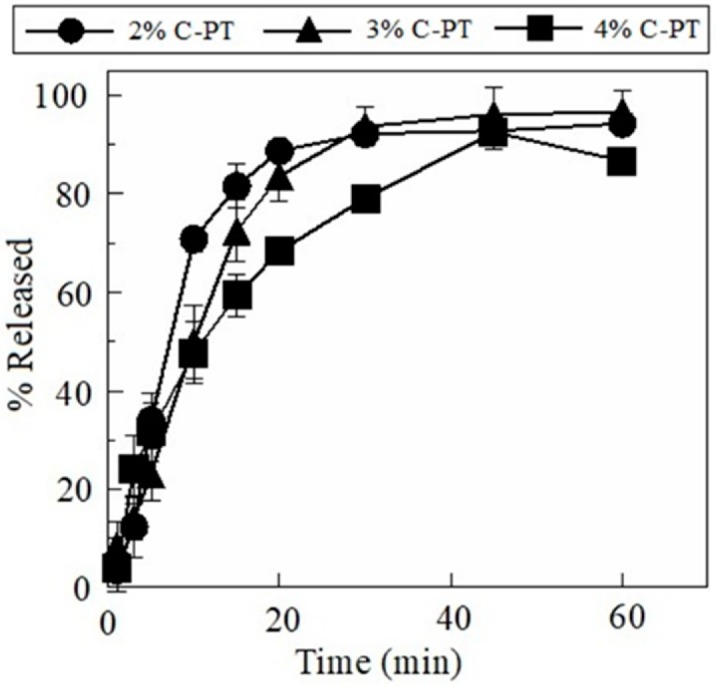
Dissolution profiles of pectin from film dosage forms prepared using citrus pectin (C-PT). Each result represents the mean and standard deviation of three determinations.

**Figure 4 materials-12-00355-f004:**
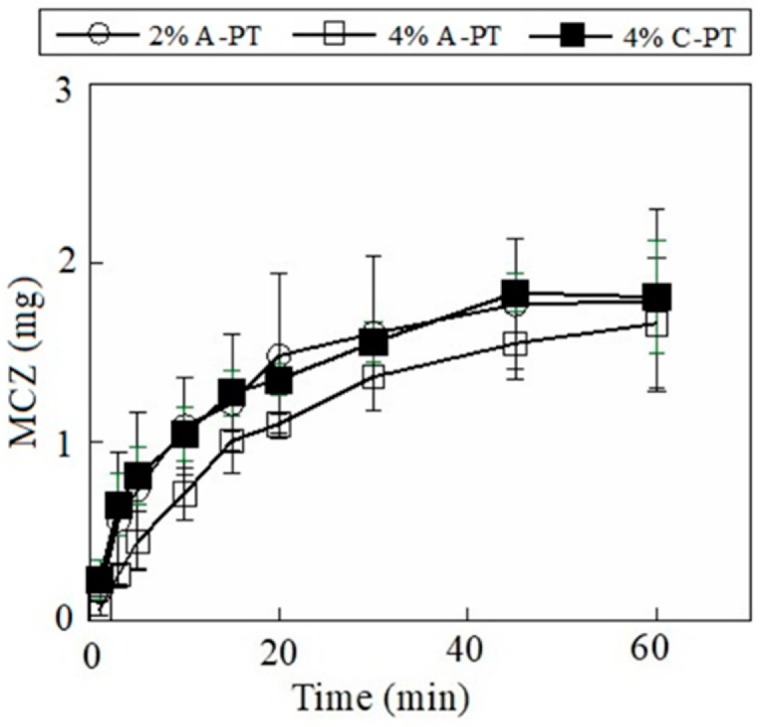
Release profiles of miconazole (MCZ) from film dosage forms prepared using pectin. Each result represents the mean and standard deviation of three determinations.

**Figure 5 materials-12-00355-f005:**
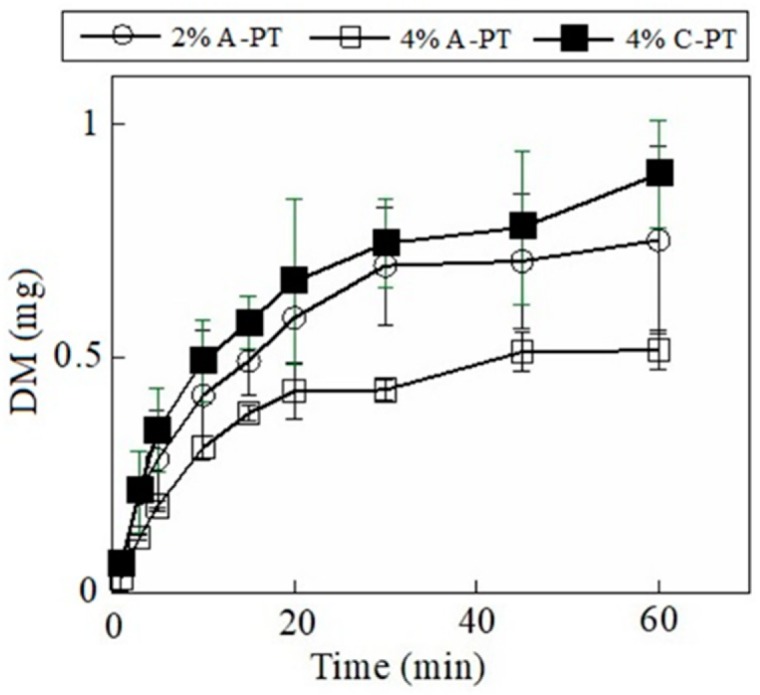
Release profiles of dexamethasone (DM) from film dosage forms prepared using pectin. Each result represents the mean and standard deviation of three determinations.

**Table 1 materials-12-00355-t001:** Viscosity of the 4% pectin solution (25 °C).

Pectin	Viscosity (mPa·s)
A-PT	500
C-PT	220
LM-5CS-J	26
LM-12CG-J	50
LM-18CG	70
X-602-03	70

(A-PT, apple pectin; C-PT, citrus pectin).

**Table 2 materials-12-00355-t002:** Thickness of film dosage forms prepared with pectin.

Pectin	Thickness (μm: mean ± SD)
2% A-PT	29 ± 1
2% C-PT	25 ± 4
3% A-PT	32 ± 2
3% C-PT	25 ± 1
4% A-PT	34 ± 2
4% C-PT	35 ± 2

Each result represents the mean and standard deviation of three determinations.

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
