# Peer review of "Drug Release Profiles and Disintegration Properties of Pectin Films"

_materials, 2019, doi:10.3390/ma12030355_

Round 1
Reviewer 1 Report
This manuscript presents the fabrication and characterization of pectin films containing model drugs. Specifically, it investigates the effect on dissolution and drug release based on the type and weight percentage of pectin solution used to cast the films. It finds that citrus pectin dissolves and releases drug more quickly than apple pectin. Overall, this study represents a very modest advancement of knowledge. The results would be easily anticipated and the writing provides no insight into (1) why the pectins might behave differently or (2) how pectin weight percentage in the initial solution affects film thickness and drug loading percent, which are likely the properties more directly responsible for the differences observed. In addition, it seems that these films are being proposed for oral delivery, but the experiments are performed in saline. Further, these films are touted as colon targeting, but dissolve in saline, which almost certainly means that they will dissolve in the mouth or stomach, which eliminates any potential targeting capabilities. Please find my additional, more specific comments below.
Major Comments:
· On lines 35-36 the authors mention that pectin is attractive because it is degraded in the colon, but is this important for something that is soluble in water? It would seem to pass through the GI tract regardless of degradation.
· Is physiological saline a representative medium to test dissolution? It would seem to be better to do the dissolution and release experiments in simulated gastric fluid and/or simulated intestinal fluid.
· In Figures 2 & 3, do the authors have an explanation for why the 60-minute time points are significantly lower than the 45-minute time points for two of the samples?
· The key difference between samples based on the weight percent used to cast the films would seem to be the resulting film thickness and drug loading weight percent, which subsequently affects dissolution and drug release; however, this is not discussed at all in the paper. Please report the thicknesses of A-PT and C-PT at 2, 3, and 4% and add a discussion of these factors to the conclusions.
· The blanket statement on line 161, “FDs are ideal dosage forms for local drug delivery” is a vast overstatement and should be removed.
Minor Comments:
· I would suggest a title change. While it might technically be correct, switching the order to “Drug release profiles and disintegration properties of pectin films” reads more clearly.
· It is up to you, but I would not abbreviate pectin “PT” because it adds confusion and does not save much space.
Author Response
Reviewer 1
We appreciate all of the helpful suggestions from you. We have examined the comments carefully and have rewritten according to your suggestions as far as possible.
Major
1. Line 35-36
As you pointed out, an important character of the film dosage form is the solubility in water. Actually, the film prepared with pectin immediately swells and disintegrates in aqueous medium. We do not propose the colonic targeting by this dosage form though we introduced the colonic targeting with pectin. We deleted the sentence “PT degraded by the microorganisms in the colon, and thus, it is a suitable material for developing a dosage form with targeted delivery to the colon [8, 9].” and added a sentence “Pectin is also an attractive material for the preparation of pharmaceuticals [new references].”
2. Film disintegration test and drug dissolution test
In this study, we tried to estimate the disintegration of film prepared with pectin in a limited aqueous medium. As you pointed out, the drug dissolution test should be practiced in simulated gastric fluid and/or simulated intestinal one if the dosage form is used orally to release the drug in gastrointestinal tract.
3. Figure 2, Figure 3
As you pointed out, 60-minute time points (2% A-PT and 4% C-PT) were lower 45-minute time points. In these time points, almost of the film dosage form disintegrated. The concentration of pectin in the test medium was determined by derivation of hydroxamic acid from pectin. We think that the coupling reaction between carboxylic acid group and hydroxylamine may be affected by characteristic of the solution such as viscosity.
4. Film thickness
We followed your advice and added the data in Table 2. The concentration of pectin did not markedly affect the film thickness and the thin film prepared with water-soluble polysaccharide quickly swelled in a limited aqueous medium. Therefore, we could not describe the relationship between the film thickness and drug release.
5. We followed your advice and removed the sentence “FDs are ideal dosage forms for local drug delivery.”.
Minor
1. We followed your advice and altered the title as “Drug Release Profiles and Disintegration Properties of Pectin Films”.
2. We followed your advice and altered “PT” to “pectin”.
Reviewer 2 Report
The authors present an interesting study about a new dosage form based on pectine films. However, despite the interest of the subject, the manuscript shows too many concerning gaps that discourage its publication in the present form.
The format and the language should be carefully revised. The text should be mainly impersonal. In addition, the presentation of the results is confusing in many parts. Figures need to be improved and figure legends should clearly inform about their content.
Results and methods do not cope. For instance, the results section starts with the characterization of the solutions mentioning the importance of the viscosity measurement. However, no information about the procedure to measure it is indicated in the methods section.
Author Response
We appreciate all of the helpful suggestions from you. We have examined the comments carefully and have rewritten according to your suggestions as far as possible.
1. Figure legends
As you pointed out, the description on each figure legend was insufficient. We added the sentence “Each result represents the mean and standard deviation of three determinations.” In Table 2 and Figures 2 - 5.
2. Results and methods, e.g. “viscosity”
We followed your advice and added a section “2.2. Viscosity of film base solution“ in Materials and Methods. Furthermore, we added Table 2 to show the results on “2.3. Preparation of FDs”.
The concentration of pectin did not markedly affect the film thickness and the thin film prepared with water-soluble polysaccharide quickly swelled in a limited aqueous medium. Therefore, we could not describe the relationship between the film thickness and drug release.
Reviewer 3 Report
Minor modifications
Row: 33, “galacturonic acid and methyl ester” I think is better like this “galacturonic acid and its methyl ester”
The introduction should be developed to include and discuss other examples of drugs prepared in the same way.
Please enhance figure 1. Is very hard to see!
Row 64. How was the drug added? As a water solution? What concentration of drug? And what was the quantity of drug solution?
Row 69. The method is not conform with those used by Pharmacopeia and I think the reader have to know this. Are this FDs prepared for a special delivery? I think the authors should have tested them according to the delivery site and I believe that the pH, the specific enzymes or tensioactive agents are very important.
The conclusions should be formulated better. The best solution is PT, in concentration of… viscosity…and so on!
I find the number of references very little!
Author Response
Reviewer 3
We appreciate all of the helpful suggestions from you. We have examined the comments carefully and have rewritten according to your suggestions as far as possible.
1. Raw. 33
We followed your advice and added “its” in the sentence.
2. Introduction
As you pointed out, film dosage form (FD) is useful for the treatment of localized problems in the oral cavity and to simplify the administration of drugs to patients. We added a sentence “The FD is a tool by which drugs can be delivered to local disease sites in the oral cavity [new references].” in Introduction.
3. Figure 1
We followed your advice and retook pictures of the films ((e) apple pectin, (f) citrus pectin), then we altered Figure 1.
4. Row 64
We used two model drugs, miconazole (MCZ) and dexamethasone (DM). Ten mg of MCZ or 2.5 mg of DM was added with agitation to 10 g of the film base solution. The mixture was thoroughly mixed with sonication, and 3.0 g of each solution was poured into individual plastic Petri dishes. In the present method, 3 mg of MCZ or 0.75 mg of DM was theoretically incorporated into each FD.
5. Row 69
As you pointed out, the drug dissolution test provided in Pharmacopeia should be practiced if the dosage form is used orally to release the drug in gastrointestinal tract. In this study, we tried to estimate the disintegration of film prepared with pectin in a limited aqueous medium. We do not propose the colonic targeting by this dosage form though we introduced the colonic targeting with pectin. Then, we deleted the sentence “PT degraded by the microorganisms in the colon, and thus, it is a suitable material for developing a dosage form with targeted delivery to the colon [8, 9].” and added a sentence “Pectin is also an attractive material for the preparation of pharmaceuticals [new references].”.
6. Conclusions
We followed your advice and added a sentence “Two pectins, A-PT and C-PT were selected as film base and 2–4% of the base solution were used to prepare FDs considering the viscosity for casting.” in Conclusions.
7. References
We followed your advice and added some references in Introduction.
Reviewer 4 Report
From the introduction, it sounds as though the films are intended for drug delivery to the colon. Something should be added to assure readers that the film wouid survive the acid environment of the stomach and arrive at the colon in tact still loaded with drug.
There is no buffer used in the dissolution and drug release studies. Dissolution of CO2 into the saline will result in a pH of abour 4 - 5. How does this compare to the colon? What about the absence of bacteria that break down PT in the colon?
In section 2.3, what is the diameter of the plastic dish?
Figures 2 - 5: The legends should indicate how many measurements are represented by each data point, what the error bars represent and what things are stastically different. A section on stastics should be aded to the Materials and Methods.
Author Response
Reviewer 4
We appreciate all of the helpful suggestions from you. We have examined the comments carefully and have rewritten according to your suggestions as far as possible.
1. Introduction, Film disintegration test and Drug dissolution test
In the present study, buffer solutions were not used in both film disintegration test and drug dissolution test. We do not propose the colonic targeting by this dosage form because the film prepared with pectin immediately swells and disintegrates in aqueous medium. However, as you pointed out, we introduced the colonic targeting with pectin. We deleted the sentence “PT degraded by the microorganisms in the colon, and thus, it is a suitable material for developing a dosage form with targeted delivery to the colon [8, 9].” and added a sentence “Pectin is also an attractive material for the preparation of pharmaceuticals [new references].”.
2. The diameter of the plastic disk
We followed your advice and described the diameter of the plastic disk.
3. Figures 2 - 5
We followed your advice and added the sentence “Each result represents the mean and standard deviation of three determinations.” in Figures 2 - 5.
Round 2
Reviewer 2 Report
The authors have made the corrections suggested and the manuscript is suitable for publication.
Author Response
We appreciate all of the helpful suggestions from you.
In the case of film prepared with sodium alginate, we tried the drug dissolution tests in both saline and phosphate-buffered saline (pH 7.2) and the similar dissolution rates were obtained though the model drug was not miconazole nitrate. In this study, we investigated not only the drug release rate but the disintegration profile of the dosage form by measuring the amount of pectin from each form that was dissolved in the test medium. However, we do not establish the exact assay of pectin in the buffered saline as yet.
As you point out, we also think that the drug dissolution test from film dosage form in buffer solution is preferable to the test in saline. Therefore, we followed your advice, and added the description “Furthermore, we think that it is necessary to investigate the drug release profile from FD in the simulated fluid at the site to be applied, such as an artificial salivary solution.” in the end of conclusion.
Reviewer 4 Report
OK
Author Response

(The authors gave the same response as above.)
